# Post-traumatic stress disorder and its associated factors among war-affected residents in Woldia town, North East Ethiopia, 2022; community based cross-sectional study

**Abenet Kassaye** [1]*, **Demeke Demilew**[2°], **Biruk Fanta**[2°], **Haregewoin Mulat**[2°], **Dawed Ali**[2°], **Jemal Seid**[1°], **Abiy Mulugeta**[3°], **Jerman Dereje**[3°]

1 Department of Psychiatry Nursing, College of Medicine and Health Science, Wollo University, Dessie, Ethiopia, 2 Department of Psychiatry, College of Medicine and Health Science, University of Gondar, Gondar, Ethiopia, 3 Department of Psychiatry, College of Health and Medical Sciences, Haramaya University, Harar, Ethiopia

☯ These authors contributed equally to this work.
* kassayeabenet@gmail.com

## Abstract

### Introduction

Post-traumatic stress disorder is marked by increased stress and anxiety following exposure to a traumatic or stressful event. Events of conflict and war-related traumas were commonly reported situations and people who have undergone through have a higher tendency to develop PTSD Woldia town had been under a serious military surge and a five-month encroachment, so the expected destruction in property, impact on physical, social and mental health of civilians was potentially high. More importantly, there is no study that investigated the significance of association between war-related traumatic events and post-traumatic stress disorder in the area. so this study aims to assess prevalence of post-traumatic stress disorder and its associated factors among war-affected residents in Woldia town, North East Ethiopia, 2022.

### Method

A community-based cross-sectional study design was employed by using a multi-stage systematic random sampling technique from May-15 to June-15/2022. A total of 609 participants were enrolled. PTSD was measured by the post-traumatic stress disorder checklist for DSM-5 (PCL-5). Data were entered by Epi data version 4.6.0.2 and analyzed using STATA version 14. Bivariable and Multivariable logistic regression analysis was done to identify associated factors to PTSD and P-values less than 0.05 were considered statistically significant.

### Results

The overall prevalence of PTSD was 56.28%. Destruction/looting of property (AOR = 1.6,95%CI,1.11–2.47), murder/injury of family member (AOR = 2.1,95% CI,1.37–3.22),

**Data Availability Statement:** The dataset used and/or analyzed during the current study are

submitted to the journal as supplementary information and can be found, in the online version, at doi.org/10.21203/rs.3.rs-2319793/v1.

**Funding:** The study was funded by University of Gondar. The funders had no role in study design, data collection and analysis, decision to publish, or preparation of the manuscript.

**Competing interests:** The authors have declared that no competing interests exist.

**Abbreviations:** AOR, Adjusted Odds Ratio; CI, Confidence Interval; DSM-5, Diagnostic and Statistical Manual 5[th] Edition; ENDF, Ethiopian National defense forces; GAD, Generalized Anxiety Disorder; HTE, Harvard trauma exposure; PCL-5, Posttraumatic stress disorder, fifth edition; PHQ-9, Patient Health Questionnaire 9 items; PTSD, Post-traumatic stress disorder; STATA, Statistics/Data Analysis.

witness of murder of family member/others (AOR = 1.6,95% CI,1.01–2.71), unlawful imprisonment (AOR = 1.7, 95%CI, 1.06–2.74), depression (AOR = 2, 95%CI, 1.37–2.93), anxiety (AOR = 3.3, 95%CI,2.26–4.97), experience trauma on themselves (AOR = 2.0,95% CI,1.22–3.58), poor (AOR = 3.1,95%CI,1.60–6.04) and moderate (AOR = 3.0, 95%CI, 1.56–5.87) social support were statistically associated with PTSD at a p-value < 0.05.

## Conclusion

The study reveals that the prevalence of PTSD was high in Woldia town following an armed conflict between Federal Government and Tigray forces. Destruction/looting of property, murder/injury of family, witness murder of family/others, unlawful imprisonment, depression, anxiety, experience on themselves, poor and moderate social support were statistically associated with PTSD. Hence, encourage organization working on mental health, routine patient assessment with a history of trauma, facilitating means to support affected residents is recommended.

## Introduction

In post-conflict and conflict-ridden societies prevalence rate of mental illness, specifically post-traumatic stress disorder can be much higher [1]. Based on DSM-5, post-traumatic stress disorder is characterized by increased stress and anxiety following exposure to traumatic event and associated with intrusive recollections of the events such as recurrent distressing memories, dreams, flashbacks, hyper-arousal symptoms of irritability and hypervigilance, problems with concentration and sleep, persistent avoidance of stimuli associated with the event and negative alternation in cognition and mood are often part of the clinical picture of the illness [2].

Our continent Africa accounts for almost 88% of the world's conflict-related morbidity and mortality as well illustrated by Virgil Hawkins's stealth conflicts map [3]. Starting from the past three decades, more than 75% of African countries have been through warfare [4] results in countless losses of lives, destruction of infrastructure and causing untold suffering for hundreds of millions of Africans and more than 70% of the casualties have been non-combatants [5]. Studies on the National and regional prevalence of posttraumatic stress disorder in Sub-Saharan Africa is estimates up to 74% and pooled prevalence of war-exposed regions was 30% [6].

The burden of untreated PTSD is enormous since it causes prolonged morbidity, impairment in day-to-day activities and poor quality of life in all dimensions including health, productivity and social interaction regardless of age, gender or socioeconomic status [7]. Based on the pooled PTSD disability ratings from the World Mental Health Survey studies, 30% of PTSD cases fall into the severe disability range [8].

In different parts of the world, the extent of post-traumatic stress disorder varies with the nature, severity, and length of traumatic events. A recent study carried out following the Russia's war operations in Ukraine and uses a convenience sample of 314 adult civilians in Ukraine revealed that 37.3% of adult Ukrainian were diagnosed with PTSD [9]. A study conducted among war affected residents in Palestinian among 139 adults exposed for political war trauma, violence at home, neighborhood and/or school showed that 53.4% of them met the diagnostic criteria for PTSD [10]. A study conducted in Syria during wartime revealed that 36.9% of adults reported full PTSD symptoms [11]. Consecutive studies were conducted on

the prevalence of PTSD among post-war population of Lebanon civilians and those with witnessed traumatic event, imprisonment, serious injury and being in combat situations in three different period quantified magnitude of PTSD ranges from 17.9% to 29.3% [12–14].

Following the 2011 Conflict in Libya, Post-conflict mental health burden were studied through population based study across different regions and PTSD prevalence was estimated at 12.4% [8]. In another study in Uganda, seven years after the conflict in three districts, (The Wayo-Nero Study) revealed, respondents with previous negative life events (familial, physical and psychological events) had prevalence of PTSD found to be 11.8% [15]. A study among victims of Boko haram terrorism in north-eastern Nigeria, more than 63% of the respondents were diagnosed with PTSD [16]. Following a three year civil war in Northern Nigeria a cross sectional study using a multistage sampling technique was study in northern Nigeria among 200 older adults, confirms 59% for full PTSD [17]. Cross-sectional study among 1200 war-affected South Sudanese adults were studied on prevalence of PTSD and founds 28% of participants with exposure to war-related traumatic events fulfilled the diagnostic criteria for PTSD [18].

In Ethiopia, PTSD among Koshe landslide survivors in Addis Ababa, shows a prevalence of 37.3% [19]. Following inter-communal violence along the borders of the Gede'o zone southern Ethiopia reveals prevalence of PTSD was 58.4% [20]. In Northwest Ethiopia, a study was conducted following Mai-kadra Massacre revealed a prevalence of PTSD was 59.8% [21].

Multiple factors contribute with the occurrence of post-traumatic stress disorder following exposure to traumatic events. sociodemographic factors like, being female [13, 19, 22], being divorced/separated or widowed [16, 19], clinical factors including depression [20], anxiety [21] and previously diagnosed for common medical disorders [14]and trauma-related factors such as having a history of exposure to traumatic events [12, 20], destruction of personal property, witness murder of family members/others [17, 20], loss of a loved one [9] and having low social support [19] are prominent and frequently reported factors in association with PTSD.

Therefore, this research was conducted in Northeast Ethiopia, Woldia town (Zonal city of North Wollo) following armed conflict between Ethiopian National defense forces(ENDF) and Tigray forces in Northern Ethiopia and later expanded to neighboring region of Amhara. Woldia town was highly affected by the war. Due to its strategic location for war, closeness to the border areas of Afar and Tigray regions, serious military surge and a five-month encroachment and several war related human right violations for relatively longer period of time than nearby towns. This study was done at a community level by using post-traumatic stress disorder checklist (PCL-5), not used previously and no published data on PTSD were done on the area. Therefore, knowing the magnitude of PTSD and its predictors in woldia town will reveal sound and reliable picture to fill information gap, used as input information to identify affected groups and add clarity to the scope of war-related mental health burdens. Moreover, this study helps the area through providing insight for stakeholders/voluntary organization to focus on post-war psychosocial rehabilitation for war affected woldia town residents.

## Materials and methods

### Study design and setting

A community-based cross-sectional study was conducted from May-15 to June 15–2022 in Woldia town, Northeast Ethiopia. Woldia town has a total population of 180,000, of whom 81,750 are men and 98,250 women. The town contains ten kebeles (the smallest administration unit in Ethiopia). The area is one of most affected area during war-brake between Federal government of Ethiopia and Tigray regional forces, leads to displacement, destruction of properties, infrastructures and injury on civilians.

## Study population

The study population includes adult residents of Woldia town and all are aged 18 and above and stays on the town for the minimum of six months.

Exclusion criteria: Individuals who/ family member/care givers reported that they were severely ill to give an interview for data collectors, those with medical certificate or related document that brief the current illness of the patient and those with observable severe physical illness/verbal communication problem were decided as not eligible for the study.

## Sample size determination and procedure

The sample size was determined by using a single population proportion by taking the prevalence of post-traumatic stress disorder (PTSD) 59.8% from a study conducted in Northwest Ethiopia, following Maikadra Massacre [21] with; 5% margin of error, 95% confidence interval as follows;

$$n = \frac{Z^2 p(1-p)}{d^2}$$

Where d = margin of error = 0.05
Z = level of confidence (95%) = 1.96
p = population proportion = 0.598

$$n = \frac{(1.96)^2 \times (0.598 \times 0.402)}{(0.05)^2}$$

N = 369.4

Since, Multi-stage sampling technique was employed, by taking account design effect, the sample size was multiplied by 1.5. So, 369.4 x 1.5 = 554, adding 10% non-response rate total number samples becomes 609.

Multi-stage systematic random sampling technique was applied to collect the required amount of sample size. Initially, four administrative unit (kebeles) was selected from the ten kebeles by lottery method and total number of households (sampling frame) in each kebele was obtained from kebele administration. Then, we allocate samples to each of Administrative unit (kebeles) proportionally depending on the total number of households under each kebele. (see Fig 1).

In a selected household, if there were more than one individual who fulfills the requirements of the study a lottery method was used. For an eligible participant who was not found at their house, the interviewers revisited the area at another time during the data collection period and if the residents were not avail during the data collection time, they were checked three times and lastly either preceding or succeeding household were nominated.

## Data collection instruments

Data was collected using interviewer administered structured and pretested questionnaires. It was collected by four BSc Psychiatry Nurses and regularly supervised by two Psychiatry professionals. The questionnaire was designed in English and translated to Amharic and back to English to maintain consistency. Data collectors were trained on how to interview participants and explain unclear questions and the purpose of the study.

PTSD was assessed by the post-traumatic stress disorder checklist (PCL-5) [23] was a 20 item self-report, measures the 20 DSM-5 symptoms of post-traumatic stress disorder. A total score is computed by adding the 20 items, so that possible scores range from 0 to 80 with a five

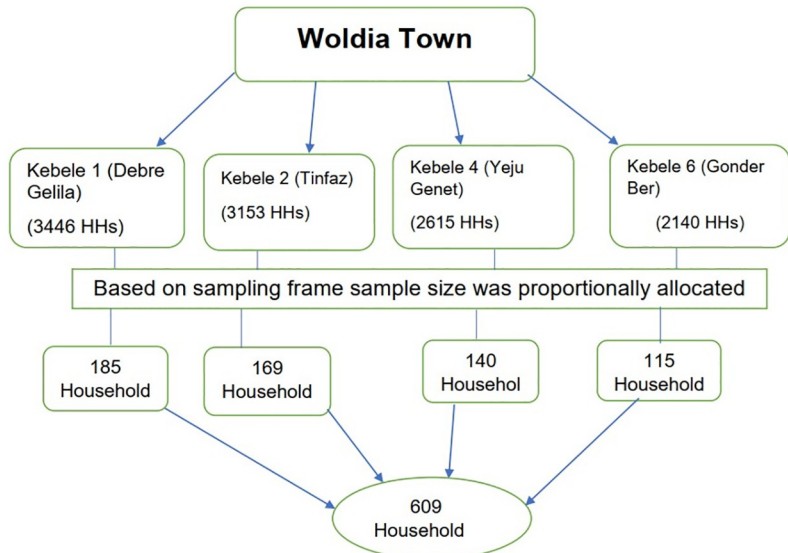

**Fig 1. Schematic presentation of sampling procedure to assessment of prevalence of post-traumatic stress disorder and its associated factors among war-affected residents in Woldia town, North East Ethiopia, 2022.**

likert scale (0 = Not at all, 1 = A little bit, 2 = Moderately, 3 = Quite a bit, 4 = Extremely) with a cut-off point of $\geq$33(PTSD). Validity and reliability of the PCL-5 had been tested and proven in a different country, for example, Zimbabwe (Cronbach's alpha = 0.92) with sensitivity and specificity of 74.5% and 70.6% respectively.

Exposure to traumatic event was assessed from the list of 16 items from Harvard trauma questionnaire (HTQ) [24] for the past one year and participants respond 'yes' for at least one traumatic event was considered experience a traumatic event. Previously adapted for use in South Sudan and has 0.87 Reliability coefficient Cronbach's $\alpha$ for the HTQ [18].

Cumulative trauma [20, 25] was assessed to examine the dose-response association between exposure to different types of traumatic events and the development of PTSD. Exposure categories were derived from various sources, including single events, two to three events, and four or more traumatic events.

Depression was measured by patient health questionnaire item nine (PHQ-9) [26], a multi-purpose instrument for screening, monitoring and measuring depression with the minimum cut-off point of 10. This is a highly reliable scale with a sensitivity of 80.8% and a specificity of 79.5% to assess depression [27].

Anxiety was measured by the GAD-7 scale [28], a seven-item instrument that is used to measure or assess generalized anxiety disorder (GAD). Each item is scored from 0 (not at all) to 4 (nearly every day). GAD-7 has adequate psychometric validity, i.e., convergent validity, diagnostic validity, factorial validity, internal consistency, test-re-test reliability in various populations.

Social support was measured using the Oslo 3-items social support scale [29] with scores ranging from 3 to 14: 3–8 = poor social support; 9–11 = moderate social support; and 12–14 = strong social support.

Socio-demographic, substance use history and clinical factors were operationalized according to different literatures [20, 21, 30].

## Data processing and analysis

The collected data were coded, entered and cleaned using Epi Data Software version 4.6.02 then it was exported and analyzed by using STATA version 14. every variable was checked in bivariate logistic regression analysis to get variables that had an association with the dependent variable, then variables with a p-value of $\leq 0.2$ were included in multivariable logistic regressions for further analysis. Finally, P-value $\leq 0.05$ in multivariate logistic regression was considered as statistically significant and the strength of associations was determined using adjusted odds ratio (AOR) at 95% CI.

## Ethical consideration and consent to participation

All procedures undertaken during data collection in accordance and ethical clearance was obtained from University of Gondar CMHS, school of medicine review committee with the ethical approval number SOM/1540/2022 and additional supportive letter was taken from department of psychiatry and taken to Woldia town Mayor Office.

Participants were informed about the aim of the study and no identification or names was recorded so as to maintain confidentiality. The study participants were informed of their right to refuse or stop participating at any time during the interview. Data collection was done by trained data collectors by visiting each selected household and for those agreed they proceed to interview by putting their signature on the informed consent sheet and for few of those participants who were not literate, data collectors read the information sheet and explained to them when necessary, and assisted them to give finger print to confirm their consent before initiating interview. The principal investigator of the study supervised the data collection process on daily base and performs a routine check on the questionnaires for consistency and completeness.

## Result

### Socio-demographic characteristics of the respondents

From a total of 609 samples, 597 participants were included in the study with a response rate of 98%. The mean age (±SD) of participants was 33.2%(±10.48) years, with a range from 18 to 81 years of age. Among participants 329(55.11%) were male and 261(43.72%) of them were in the age range of (25–34) (see Table 1).

### Trauma exposure of participants

The majority of participants 460(77.05%) reported exposure to Noise of explosions during wartime, 208(34.84%) participants lost/face serious injury to their family members, friends or others important and 129(21.61%) individuals report witnesses of murder of either of their family member, friend or unknown others. In terms of cumulative traumatic events, 43.4% of respondents had experienced two to three traumatic events, and 92 (15.4%) out of 238 participants had been exposed to four or more traumatic events (see Table 2).

### Clinical, social support and substance related factors

Out of the total participants, 47(7.87%) of them were found to have previous history of mental illness and half 315(52.76%) participants were positive for depression and 259(43.38%) of them were positive for anxiety. On social support, 284(47.57%) had poor social support and 73 (12.23%) had a strong social support. Regarding substance-related factors, 287 (48.07%) and 125(20.94%) of respondents had ever used history for alcohol and khat respectively. Regarding current use of a substance, 240(40.2%) of them currently use alcohol.

**Table 1. Socio-demographic characteristics among war-affected residents in Woldia town, North East Ethiopia, 2022 (n = 597).**

| Variable | Category | Frequency n = 597 | Percentage (%) |
|---|---|---|---|
| Sex | Male | 329 | 55.11 |
| | Female | 268 | 44.89 |
| Age (years) | 18–24 | 110 | 18.43 |
| | 25–34 | 261 | 43.72 |
| | 35–44 | 137 | 22.95 |
| | 45–54 | 62 | 10.39 |
| | 55–64 | 19 | 3.18 |
| | Above 65 | 8 | 1.34 |
| Religion | Orthodox | 273 | 45.73 |
| | Muslim | 215 | 36.01 |
| | Protestant | 64 | 10.72 |
| | Catholic | 43 | 7.20 |
| | Other | 2 | 0.34 |
| Marital status | Single | 176 | 29.48 |
| | Married | 353 | 59.13 |
| | Divorce | 44 | 7.37 |
| | Widowed | 24 | 4.02 |
| Having children | Yes | 333 | 55.78 |
| | No | 264 | 44.22 |
| Educational status | Can't read & write | 43 | 7.20 |
| | Primary school | 79 | 13.23 |
| | Secondary school | 134 | 22.45 |
| | Diploma | 177 | 29.65 |
| | Degree & above | 164 | 27.47 |
| Occupation | House-wife | 51 | 8.54 |
| | Government employee | 211 | 35.34 |
| | Private employee | 103 | 17.25 |
| | Daily laborer | 44 | 7.37 |
| | Self-work/private | 149 | 24.96 |
| | Students | 39 | 6.54 |

* Other Religion = Adventist

## Prevalence of post-traumatic stress disorder

The study reveals that, prevalence of post-traumatic stress disorder (PTSD) was 56.28% with (95% CI, 52.2, 60.2). The estimated prevalence rate was 53.6% and 46.4% among males and females respectively.

## Factors associated with post-traumatic stress disorder

In Bivariable logistic regression analysis variables like age group, experience the traumatic event on themselves, destruction/looting of personal property, lack of food and water, murder/injury of a family member /friend, a witness of the murder of family member /friend/other, made to accept ideas against will, unlawful imprisonment, poor and moderate social support, depression and anxiety were to have a p-value of less than 0.2 and these variables fulfills the requirements for further analysis in multivariable logistic regression analysis.

**Table 2. Description of exposure to traumatic events and way of exposure among war-affected residents in Woldia town.**

| Variable | Category | Male | Female | Total n = 597 | Percentage (%) |
|---|---|---|---|---|---|
| Exposure to traumatic event | Destruction/looting of personal property | 172 | 123 | 295 | 49.41 |
| | Noise of explosion | 248 | 212 | 460 | 77.05 |
| | Lack of food or water | 226 | 187 | 413 | 69.18 |
| | Murder/injury of family member /friend | 125 | 83 | 208 | 34.84 |
| | Witnessing the murder of family/friend/other | 73 | 56 | 129 | 21.61 |
| | Ill health without medical care | 129 | 126 | 255 | 42.71 |
| | Made to accept ideas against will | 78 | 94 | 172 | 28.81 |
| | Physical trauma/ Serious injury | 78 | 76 | 154 | 25.80 |
| | Unlawful Imprisonment | 79 | 72 | 151 | 25.29 |
| Cumulative trauma (Exposure to different traumatic events) | 1 | 148 | 98 | 246 | 41.2 |
| | 2–3 | 131 | 128 | 259 | 43.4 |
| | 4 and above | 50 | 42 | 92 | 15.4 |
| Way of exposure | It happened to me | 161 | 129 | 290 | 48.58 |
| | Directly I witnessed it | 97 | 70 | 167 | 27.97 |
| | I learned about it | 54 | 51 | 105 | 17.59 |
| | Exposed as part of my job | 17 | 18 | 35 | 5.86 |

In multivariable logistic regression analysis, factors Destruction/looting of property, murder/ serious injury of a family member/friend or others important, a witness of the murder of family member/ friend/others, unlawful imprisonment, experience traumatic event on themselves and having a poor and moderate level of social support, depression and anxiety were statically significant for post-traumatic stress disorder with p-value less than 0.05.

The odds of developing post-traumatic stress disorder were 1.6 times higher among participants with the destruction/looting of their properties as compared to their counterparts (AOR = 1.6, (1.11, 2.47)). the odds of having PTSD among those with murder/ injury of their family/friend or other had two-fold higher as compared to those who didn't have (AOR = 2.1, (1.37, 3.22)) and the odds of developing PTSD among individuals witnessing murder was 1.6 times higher as compared with those didn't witness (AOR = 1.6, (1.01, 2.71)). Individuals with unjudicial imprisonment against their will had odds of 1.7 times higher as compared to their counterparts (AOR = 1.7, (1.06, 2.74)). In comparing way of exposure to traumatic event, individuals exposed to traumatic events through learning, the odds of having PTSD among those experience on themselves is twofold higher (AOR = 2.0, (1.22, 3.58)).

The odds of developing PTSD were more than threefold in participants with anxiety as compared to their counterparts (AOR = 3.3, (2.26, 4.97)) and for those with depression, the odds of having PTSD were two times higher in compared to their counterparts (AOR = 2.0, (1.37, 2.93)).

As compared to those with strong social support, those with poor social support have an odd of 3.1 times and those with moderate social support have odds of 3.0 times higher for developing PTSD (AOR = 3.1, (1.60, 6.04)) and (AOR = 3.0, (1.56, 5.87)) respectively (see Table 3).

**Table 3. Bivariate and multivariate logistic regression analysis of selected factors and post-traumatic stress disorder(PTSD) among war-affected residents in Woldia town, North East Ethiopia, 2022 (n = 597).**

| Explanatory variables | Post-traumatic stress disorder(PTSD) | | | | | |
|---|---|---|---|---|---|---|
| | Yes | No | P-value | COR | P-value | AOR with CI |
| Age group | | | | | | |
| 18–24 | 53 | 57 | 1 | 1 | 1 | 1 |
| 25–34 | 137 | 124 | 0.448 | 1.1(0.71, 1.78) | 0.641 | 1.1(0.67, 1.90) |
| 35–44 | 94 | 43 | 0.001 | 2.3(1.38, 3.97) | 0.114 | 1.6(0.89, 2.96) |
| 45–54 | 35 | 27 | 0.298 | 1.3(0.73, 2.58) | 0.627 | 0.8(0.40, 1.71) |
| 55–64 | 11 | 8 | 0.436 | 1.4(0.48, 3.61) | 0.940 | 0.9(0.32, 2.81) |
| Above 65 | 6 | 2 | 0.162 | 3.2(0.61, 16.4) | 0.561 | 1.6(0.29, 9.54) |
| Shortage of food or water | | | | | | |
| Yes | 245 | 168 | 0.025 | 1.4(1.05, 2.11) | 0.601 | 1.1(0.73, 1.69) |
| No | 91 | 93 | 1 | 1 | 1 | 1 |
| Murder/injury of family | | | | | | |
| Yes | 143 | 65 | 0.001 | 2.2(1.56, 3.18) | 0.001 | 2.1(1.37, 3.22)** |
| No | 193 | 196 | 1 | 1 | 1 | 1 |
| Witnessing murder of family/friend/others | | | | | | |
| Yes | 92 | 37 | 0.001 | 2.2(1.56, 3.18) | 0.047 | 1.6(1.01, 2.71) * |
| No | 244 | 224 | 1 | 1 | 1 | 1 |
| Destruction/looting of property | | | | | | |
| Yes | 188 | 107 | 0.001 | 1.8(1.31, 2.53) | 0.012 | 1.6(1.11, 2.47) * |
| No | 148 | 154 | 1 | 1 | 1 | 1 |
| Made to accept ideas against will | | | | | | |
| Yes | 104 | 68 | 0.169 | 1.2(0.88, 1.82) | 0.321 | 0.8(0.51, 1.24) |
| No | 232 | 193 | 1 | 1 | 1 | 1 |
| Unlawful imprisonment | | | | | | |
| Yes | 105 | 46 | 0.001 | 2.1(1.43, 3.14) | 0.025 | 1.7(1.06, 2.74) * |
| No | 231 | 215 | 1 | 1 | 1 | 1 |
| How you experience it? | | | | | | |
| It happened to me | 181 | 109 | 0.001 | 2.3(1.51, 3.77) | 0.007 | 2.0(1.22, 3.58) * |
| Directly I witnessed it | 92 | 75 | 0.024 | 1.7(1.07, 2.89) | 0.743 | 1.1(0.61, 1.96) |
| I learned about it | 43 | 62 | 1 | 1 | 1 | 1 |
| Exposed as part of my job | 20 | 15 | 0.098 | 1.9(0.88, 4.16) | 0.132 | 1.9(0.81, 4.88) |
| Anxiety | | | | | | |
| Yes | 192 | 67 | 0.001 | 3.8(2.71, 5.48) | 0.001 | 3.3(2.26, 4.97) ** |
| No | 144 | 194 | 1 | 1 | 1 | 1 |
| Depression | | | | | | |
| Yes | 215 | 100 | 0.001 | 2.8(2.04, 3.99) | 0.001 | 2.0(1.37, 2.93) ** |
| No | 121 | 161 | 1 | 1 | 1 | 1 |
| Social Support | | | | | | |
| Poor | 177 | 107 | 0.001 | 5.0(2.81, 9.06) | 0.001 | 3.1(1.60, 6.04) * |
| Moderate | 141 | 99 | 0.001 | 4.3(2.40, 7.85) | 0.001 | 3.0(1.56, 5.87) * |
| Strong | 18 | 55 | 1 | 1 | 1 | 1 |

*statistically significant

* = p<0.05, and ** = p<0.001

Chi square = 7.16, Hosmer-lemshow test = 0.52

## Discussion

The study quantifies that, the prevalence of post-traumatic stress disorder (PTSD) was 56.28% with (95% CI, 52.2, 60.2) among residents of Woldia town and this finding goes in line with the previous studies conducted following the Mai-kadra massacre, Northwest Ethiopia 59.8% [21], Gede'o zone, Southern Ethiopia 58.4% [20], Northern Nigeria 59% [17] and Northern Uganda 57% [31].

While comparing, the result of this study was lower than other studies conducted in North-Eastern Nigeria 63.7% [16], and Siri Lanka 68% [27]. The possible justification could be, variation in the study population, this study includes Woldia town residents aged 18 and above, but in a study conducted in, North-Eastern Nigeria and Siri Lanka participants were only 15 to 35 years, school-aged, children and adolescents respectively.

Another reason for the variation could be the time gap from a traumatic event to the study, in North-Eastern Nigeria a study was conducted among Internally displaced populations after six weeks and as the time of study is close to the event participants might have a high potential to be symptomatic. Also existing geographical and socio-cultural variations contribute for the differences.

On the other side, the finding of this research is higher than other studies conducted in South Sudan 28% [18], Northern Uganda(The Wayo-Nero Study)11.8% [15], Ukraine 37.3% [9] and southern Lebanon 29.3% [13, 32]. The discrepancy was due to variation in the assessment tool, in South Sudan MINI International Neuropsychiatric Interview (MINI) was employed and in Southern Lebanon, PTSD was assessed by checklist from DSM-IV-TR PTSD symptom criteria. In contrast, this study utilizes post-traumatic stress disorder checklist (PCL-5 with extended Criterion A of DSM-5) assessment tool with a better internal consistency to measure PTSD [20].

Another reason is due to the difference in psychometric properties of tools, a study conducted in South Lebanon shows a sensitivity of 78% [14], but the tool we use has a sensitivity of 81.55% results more detection of true positive cases, so increases the prevalence of PTSD.

More than 66%(398) of participants were in the age range between 25–44 and among those identified positive for PTSD 68% of them are members of this age group. Those in this age range are at imminent risk for exposure to war-related traumas since adults are at the right age to join military camps and participate in combat fields [33].

Destruction/looting of personal property was 1.6 times higher to develop PTSD agreed with studies conducted in Gede'o zone, Ethiopia [20] and South Sudan and Liberia [34] such devastating situations are shocking, difficult to accept, adjust self to previous and puts individuals and family future life at risk and acute stress is a typical response [35].

Participants who lost their family member/friend were twice as likely to have PTSD, it was supported by studies done in Northern Nigeria [17] and Mai-kadra, Ethiopia [21], this is due to their chance of exposure to traumas in some extent and perception of subsequent injury and distress for later traumas, also had negative intrusive thought of revenge, payback and impact on emotional well-being of victims family members [20].

Those study participants who had a direct witness to murder of family members/ others were 1.6 times more likely to have PTSD, agreed to studies conducted in South Sudan and Liberia [34], and Gede'o zone, Ethiopia [20], This is due to traumatic events especially direct witnessing having a higher tendency to be captured in mind and later reprocessed in the form of nightmares, flashbacks, hyperarousal, psychological distress and later PTSD [19, 21].

Individuals, who pass through unlawful imprisonment had 1.7 times higher to develop PTSD as supported by studies conducted in Northern Nigeria [17], It might be due to the absence of law enforcement bodies and this leads way for mal treatment of civilians,

threatening, manipulation, physical abuse, torture and being secluded from families and social contact increase the likelihood of trauma exposure and leads to mental distress and PTSD [35].

Participants with anxiety were 3.3 timed higher odds of developing PTSD as compared with others didn't have, as supported by other studies done in Turkey [36], northern Cameron [33] and Mai-kadra, Ethiopia [21]. Even if, casual relation between anxiety and PTSD couldn't concluded, comorbid cases of anxiety disorders make persons more vulnerable to develop PTSD [35]. Additionally, it was discovered that anxiety symptoms could independently predict PTSD and there was a strong association between exposure to trauma and anxiety symptoms [20].

The odds PTSD were twofold higher among respondents with depression as supported by studies conducted in Southern Lebanon [12], Gede'o zone and Addis Ababa, Ethiopia [20, 22], the onset of depression is associated with preexisting vulnerabilities and subsequent psychosocial stressors, whereas the onset of PTSD is associated with stressors related to disaster-related events [14] and up to 86% develops co-occurrence [37]. Also level of depressive symptoms was high as the level of exposure increased and likelihood of having comorbid disorders of PTSD and depression was elevated following war related traumatic events [18, 31] and association of symptoms between PTSD and depression was observed [15].

Participants with poor and moderate social support had 3.1 and 3times higher odds to develop PTSD as compared to those with strong social support. This finding also goes along with other studies conducted in South Lebanon [14], Turkey [36] and Koshe landslide survivors [19]. This is possibly due to the fact that poor mental health might result from traumatic injury and peoples with strong social support allows people to express their worries, feel secure, and have a sense of belonging, social networks and strong social support systems may be able to lessen the impact of stressful life events.

Individuals who directly experience the traumatic event on themselves had two times higher odds of developing PTSD as compared with those learning the traumatic event from close family or close friend, this finding agrees with studies conducted in Netherlands [38] and Northern Uganda [31]. This is because, direct experiences have physical, psychological and emotional effects comparable to and possibly worse than those of witnesses or other forms of exposure [35].

## Conclusion

Since the study was conducted in a short period after armed conflict in the area and beyond the infrastructural destruction and physical disability, psychological and mental health-related crisis is potentially high, as this study quantifies, more than 50% of participants develop post-traumatic stress disorder(PTSD). Especially, individuals on whom trauma happens, witnessing trauma in person, murder/injury of a family member /friend, destruction/looting of property, unlawful imprisonment, depression, anxiety, poor and moderate social support were significantly associated with post-traumatic stress disorder(PTSD). The finding implies the association of exposure to different war-related traumatic events for the development of PTSD, lends a strong theoretical and practical significance for considering strengthen mental health care services for the community. Furthermore, depression and anxiety disorders contribute as a factor for PTSD and this finding necessitates a planned intervention and making informed clinical assessment of PTSD and possible co-morbidity and renders psychiatric and post-conflict psychosocial rehabilitation in the community. The association between poor and moderate level of social support for PTSD needs the strengthening and improvement of social network of the community. Specifically, by using existing self/others help programs, then

engaging community leaders in the social gathering and facilitate in restoration of their relationship and interaction.

## Strength and limitation of the study

As a strength, such study is not done previously in the area and its timely and reveals the current impact of war on the community and done by adding important factors not included in previous studies and applies a standardized tool, PCL-5, post-traumatic stress disorder checklist (PCL-5) and Harvard trauma questioner(HTQ) not used previously. As, a drawback the utilization of items that needs remembering of previous events in the questionnaire may invite recall bias and we were not able to verify whether the depressive and anxiety symptoms preceded or followed the PTSD. Sensitive traumatic events like rape and sexual violence is not assessed because of social desirability issue and under reporting and we strongly encourage to conduct such focused studies through different methodologies.

## Supporting information

**S1 Data.**
(XLS)

## Acknowledgments

We would like to express my acknowledgment for all of study participants, spending their time and willingness to share their experience with full trust. We would like to thank to Woldia town administration mayor Mr. Gize Degu Kassaw for their reception and permission as well as data collectors for their commitment and eager participation.

## Author Contributions

**Supervision:** Demeke Demilew, Biruk Fanta, Haregewoin Mulat, Dawed Ali, Jemal Seid, Abiy Mulugeta, Jerman Dereje.

**Writing – original draft:** Abenet Kassaye.

**Writing – review & editing:** Abenet Kassaye.

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
