## [Decision Letter · Decision Letter 0]

22 May 2023

PONE-D-22-34183POST-TRAUMATIC STRESS DISORDER AND ITS ASSOCIATED FACTORS AMONG WAR-AFFECTED RESIDENTS IN WOLDIA TOWN, NORTH EAST ETHIOPIA, 2022; COMMUNITY BASED CROSS-SECTIONAL STUDY.PLOS ONE

Dear Dr. kassaye,

Thank you for submitting your manuscript to PLOS ONE. After careful consideration, we feel that it has merit but does not fully meet PLOS ONE’s publication criteria as it currently stands. Therefore, we invite you to submit a revised version of the manuscript that addresses the points raised during the review process.

Both reviewers stated that your manuscript can be meaningful contribution to the literature and suggested ways to improve your manuscript.

In particular, reviewer 2 pointed to important aspects that needs to be addressed carefully before a final decision on your manuscript can be made. I kindly request you addressing each of the points raised by the reviewers and revise the concerning parts of your manuscript accordingly.

We look forward to receiving your revised manuscript.

Kind regards,

Inga Schalinski

Academic Editor

PLOS ONE

Journal Requirements:

Additional Editor Comments (if provided):

Please revise the references style according to the PLOS ONE submission guidelines.

Ethics: Please provide information how the informed consent was retrieved from the participants.

The language needs to be improved and typos removed.

In the introduction; please summarize the current state of literature regarding the risk of specific exposures to traumatic events. Please use the existing literature to relate to your research questions, instead of listing the heterogenous results on PTSD rates across different countries.

The co-occurrence of various trauma event types is high in war-affected area and should be reported and addressed in the results section.

Minors:

Correct all concerning p-values to p <.001 (instead of p = 0.000)

Please evaluate all directions of comorbidity when discussion the high co-occurrence of disorders:

-Common Cause: Trauma predispose for depression and PTSD

-PTSD results in Depression

-And Previous Depression renders the risk for individuals to develop PTSD…

Reviewers' comments:

Reviewer's Responses to Questions

**Comments to the Author**

1. Is the manuscript technically sound, and do the data support the conclusions?

Reviewer #1: Yes

Reviewer #2: Yes

2. Has the statistical analysis been performed appropriately and rigorously? 

Reviewer #1: Yes

Reviewer #2: Yes

3. Have the authors made all data underlying the findings in their manuscript fully available?

Reviewer #1: Yes

Reviewer #2: No

4. Is the manuscript presented in an intelligible fashion and written in standard English?

Reviewer #1: Yes

Reviewer #2: No

5. Review Comments to the Author

Reviewer #1: 1. In the study design and setting, it is better to cancel the sentence "The study aims to assess prevalence of post-traumatic stress disorder (PTSD) and its associated factors among war-affected residents in Woldia town".

2. You better to rewrite the exclusion criteria.

3. Hence you have used the multistage sampling technique, why you did not consider the design effect while you calculate the sample size? Better to support the multistage in Figure(s).

4. It is better if you tried to combine the strengths and limitations in one paragraph.

5. you better to acknowledge your participants rather than your co-authors.

Reviewer #2: The manuscript addresses the prevalence of PTSD in Woldia town, Ethiopia. Especially in regions with severe conflicts and war, it is important to address the mental health of the residents to provide appropriate support services.

There is a general comment which is not addressed concerning the overall benefit of the manuscript. It is not clear why the authors choose especially that town because there is one existing article of another town in Northwest Ethiopia. It should be more clear why it is important to know the prevalance of another town and what the implications of these results are (e.g. regarding support services, practical implications).

In addition, the focus should also be more on PTSD subscales and subgroups (e.g. sex, educational status). Means and SD should be presented for PTSD and subscales and maybe also specific sociodemographic subgroups. At this point, the picture of the distribution of PTSD is too vague.

Furthermore, there are at the moment major revisions needed:

For publication, the manuscript needs a linguistic revision.

Introduction

In general, it is not quite clear why the authors choose especially this town – because there is at least one other publication from a city in Northwest Ethiopia. The introduction is lacking a narrative of why it is important to know this specific prevalence rate and what gap will be filled with these results. It would be great, if the authors could stress this in more detail.

Line 61-64. The text says (line 64): “…are often part of the clinical picture of the illness”. Here, the authors stated the diagnostic criteria of PTSD regarding DSM and therefore, all the criteria mentioned must be part of the clinical picture of PTSD. This should be stated much clearer.

Line 81-112: the authors name a lot of different studies, but it is not necessary to list all of them in such detail. It would be better for the reader if the authors cite prevalence rates from meta-analyses or reviews, e.g., there are some regarding PTSD in conflict-related regions worldwide and also regarding PTSD in the MENA region. After stating these more global numbers, it would be perfect to name single prevalence studies from either regions similar to the situation in Woldia or others from Ethiopia. That depends on the aim of the authors. It might also be a benefit if the authors focus only on adults in their literature

Line 113-119: for the reader it would be easier to cluster all these factors in sociodemographic, trauma-related and psychopathological factors.

Materials and Methods

Line 141/142: How did the authors assess and decide if a person is to severely sick or unable to communicate?

Could you please insert the Figure 1 in the text to have a reference in the text for the figure. That would be very helpful.

Line 164: it would be better to set the citation for the questionnaire directly after the first mention, i.e. (PLC-5, “citation of author of the questionnaire”).

Line 172: the same for the HTQ. It would be easier to identify the source if it directly follows naming the questionnaire (HTQ, "source").

The same for Line 175, Line 179, line 184.

Line 184: Is the questionnaire (Oslo 3-item social support scale) originally quoted in the mentioned literature (34)?

Line 187-188: the authors stated that “sociodemographic, substance use history and clinical factors were operationalized according to different literature”. The (35) literature refers primarily to alcohol and drug abuse. Is this source also refering to other clinical factors?

Line 191-193: I do not fully understand why you first checked variables in a bivariate logistic regression and then enter them in a multivariate logistic regression? What is the rationale behind this? The second question refers to the p-value of <0.2. Why do you use this p-value?

Results

In general, the authors could decide which information they want to state in the text. At the moment, every result in the text has its counterpart in a table. Maybe there is some space to shorten the text.

Table 1:

a minor comment: please add “years” in the column for “Age”.

It is very interesting that 35.34% of the participants are government employees as well as that 55% have children. are these typical numbers for Woldia? It would be great if you can say something in the discussion.

Table 2: it is not quite clear why the authors split the variable “way of exposure” into male and female but not the variable “exposure to TE”.

In general, for a better understanding it would be helpful to state that the “way of exposure” refers to the most traumatic event but the variable “exposure to traumatic event” is not.

it is not quite clear why the authors only list certain traumatic events but not all. It would be helpful for the reader to have an overview of the frequency of all traumatic events (it could also be in the supplement section).

Line 225-231. It would be great to also have the mean scores (and SD) for depression, anxiety and social support.

Line 233-235. It is important to also have the mean score (and SD) for PTSD, the means for the clusters and maybe also all these means with regard to different sociodemographic variables (e.g. sex: male/female, education). The paper deals with PTSD and therefore, it must have a greater focus on this variable and I think it is not sufficient to only report the results of the regression analysis but to have a closer look on PTSD means in different subgroups.

Line 236-241: I have to ask under which assumptions only these named variables fulfil minimum requirements for further analysis. To me, it is not clear and it would be helpful for the reader to better understand these requirements. In addition, the p value less than 0.2 should also be explained.

With regard to the different traumatic events, I am curious why sexual assault seems to be not relevant in this context. Do you have an explanation?

Lines 248 ff: Since the individual categories being compared here are not exclusive (i.e., a person who experienced x may have experienced y as well), I think it might be difficult to make these comparisons. It is clear from Table 1 that ca. 50% have experienced looting of personal property and ca 35% have also experienced murder/injury of a family member/friend. These persons can be completely overlapping. Was this considered in the calculation?

Especially when looking at table 3, it shows that nearly half of the persons who did not experience the specific traumatic events also were diagnosed with PTSD.

Discussion

In general, it would be great to include more clinical implication.

Line 274-297. It does not have to be so detailed when it comes to the classification of the prevalence rate. I think it would be sufficient to state 1-2 sources with similar background and state that there are also other rates with a similar population but that different reasons have an influence on the rates.

Line 308: I am not sure if the authors want to refer to acute stress as a response to destruction/looting or to PTSD.

Line 312-313: did the authors ask about the examples of the intrusive thoughts or are these assumptions they made due to the experience of traumatic events?

Line 318: a source is needed for the stated fact.

Line 361-363. There are other limitations to this study, e.g. that PTSD

6. PLOS authors have the option to publish the peer review history of their article (what does this mean?). If published, this will include your full peer review and any attached files.

Reviewer #1: **Yes: **Agmas Wassie Abate

Reviewer #2: No

---

## [Author Response · Author response to Decision Letter 0]

13 Jul 2023

I would like to express my deep-felt appreciation for the substantial time, effort you devote on this paper and opportunities you gave me to respond over the raised questions. As I was a junior researcher, I am admired by your deep inspection, point of view and constructive comments. 

Thanks Again.

---

## [Decision Letter · Decision Letter 1]

16 Aug 2023

PONE-D-22-34183R1POST-TRAUMATIC STRESS DISORDER AND ITS ASSOCIATED FACTORS AMONG WAR-AFFECTED RESIDENTS IN WOLDIA TOWN, NORTH EAST ETHIOPIA, 2022; COMMUNITY BASED CROSS-SECTIONAL STUDY.

PLOS ONE

Dear Dr. kassaye,

Thank you for submitting your manuscript to PLOS ONE. After careful consideration, we feel that it has merit but does not fully meet PLOS ONE’s publication criteria as it currently stands. Therefore, we invite you to submit a revised version of the manuscript that addresses the points raised during the review process.

Thank you for sharing the revised version of the manuscript. I appreciate the significant improvements you've made. However, to further enhance the paper's quality, it would be beneficial to address the remaining aspects highlighted by reviewer 2 and incorporate my own review suggestions. I kindly request a point-by-point response to these issues.

Furthermore, please ensure consistency in capitalization throughout the manuscript, checking for proper use of title case for headings and subheadings, and appropriate lower case for regular text. This will ensure the overall professionalism and readability of the paper. Thank you for your attention to these details, and I look forward to reviewing the updated version.

We look forward to receiving your revised manuscript.

Kind regards,

Inga Schalinski

Academic Editor

PLOS ONE

Journal Requirements:

Additional Editor Comments:

Please be clear about your cumulative measure of cumulative trauma (= exposure to the number of different traumatic events). This score should be defined in the method section and also the rational for the categories. Do you find a correlation when using the continuous cumulative trauma score and severity of PTSD (dose-response relationship)?

It is quite surprising that there is no cumulative exposure effect in your sample as it has been reported in previous studies from war affected populations (Neuner et al., 2004). This result should be discussed thoughtfully.

Please clarify the “way of exposure“. Is this type of exposure related to the worst event?

You may want to consider another recent study about the importance of sexual violence in North East Ethipia (Tenaw, L. A., Aragie, M. W., Ayele, A. D., Kokeb, T., & Yimer, N. B. (2022). Medical and psychological consequences of rape among survivors during armed conflicts in northeast Ethiopia. PLoS one, 17(12), e0278859.)

Furthermore; please use references for the fact of social desirability and underreporting of sexual violence. It may be useful to encourage future studies to assess exposure to sexual violence.

In table 3: Please clarify why you have put some numbers in bold.

In the abstract: “in the area. so this study”

Please be careful about large and lower cases

Or in the discussion: “events. sociodemographic factor”

Or in the introduction: “Therefore, Knowing”

Or in the introduction: “in woldia town”

Or in the results “. the odds of having PTSD among”

Or in the methods: “STATA version 14. every variable”

Neuner, F., Schauer, M., Karunakara, U., Klaschik, C., Robert, C., & Elbert, T. (2004). Psychological trauma and evidence for enhanced vulnerability for posttraumatic stress disorder through previous trauma among West Nile refugees. BMC psychiatry, 4(1), 1-7.

Reviewers' comments:

Reviewer's Responses to Questions

**Comments to the Author**

1. If the authors have adequately addressed your comments raised in a previous round of review and you feel that this manuscript is now acceptable for publication, you may indicate that here to bypass the “Comments to the Author” section, enter your conflict of interest statement in the “Confidential to Editor” section, and submit your "Accept" recommendation.

Reviewer #1: All comments have been addressed

Reviewer #2: (No Response)

2. Is the manuscript technically sound, and do the data support the conclusions?

Reviewer #1: Yes

Reviewer #2: Yes

3. Has the statistical analysis been performed appropriately and rigorously? 

Reviewer #1: I Don't Know

Reviewer #2: Yes

4. Have the authors made all data underlying the findings in their manuscript fully available?

Reviewer #1: Yes

Reviewer #2: Yes

5. Is the manuscript presented in an intelligible fashion and written in standard English?

Reviewer #1: Yes

Reviewer #2: Yes

6. Review Comments to the Author

Reviewer #1: I recommend to the authors to send a point by point response for each questions that has been raised in the previous round.

Reviewer #2: Thank you for your revised manuscript which is well improved.

there are some comments left which I would like to adress. For this, I used the same numeration as you did for your revision, therefore, the numbers match your numbers in the revision.

General:

#1 Thank you for including more about the rationale behind the paper. In your response you wrote: “The rationale behind choosing Woldia town for PTSD assessment was the scope of armed conflict. During the armed conflict, Woldia town was highly affected demographically by the war, due to its strategic location for war, closeness to the border areas of Afar and Tigray regions, serious military surge and occupation for more than five months and several war related human right violations for relatively longer period of time than nearby towns. Therefore, Knowing the magnitude of PTSD and its predictors in woldia town will reveal sound and reliable picture to fill information gap. Moreover, this study used as input to identify affected groups, add clarity to the scope of war-related mental health burdens and helps the area through providing insight for stakeholders/voluntary organization to focus on post-war psychosocial rehabilitation for war affected woldia town residents”.

In the Introduction section, you only used some parts of the explanation, but I find it important to include the whole explanation as you send it.

#2 Thank you for your explanation regarding the binary outcome of PTSD. Nevertheless, I find it important to have an idea about the severity of PTSD (knowing if the mean is near the cut-off or way above) and it would be great if you can include the M and SD for PTSD additionally to the binary outcome.

Introduction

#2 Could you additionally include that you refer to the DSM-5 PTSD criteria. That would be helpful to know it right away without looking at the reference. Thank you.

#3 Thank you for including global figures and other surveys. This improved the introduction. For a more focused narrative, I would suggest deleting the studies from Ukraine, Syria and Lebanon because you refer to so many other countries in Africa that this already makes your point. To refer to the WHO global survey and here to different African countries and in addition to individual other studies from Africa is sufficient.

In addition, I would like to encourage you to add more details in the section where you address contributing factors to the occurrence of PTSD (ll 121-128) if they are from studies conducted in Africa.

#5 Thank you for explaining the criteria regarding “severely sick” or “unable to communicate”. It would be helpful if you include this explanation also in the manuscript.

#11 and #18 Thank you for the clarification. Nevertheless, it would be great to know why the p-value is <.2 and <.25. I do not fully understand the decision behind the p-values.

Additional new comments:

Table 2: please add the frequencies for male and female in the columns. At the moment you report the sample size for male and female but due to different sizes, it is hard to compare both columns. You already have the frequency for the total sample and it would be helpful to also have it for male/female.

Table 3: please include the abbreviation AOR and CI in the notes below the table.

7. PLOS authors have the option to publish the peer review history of their article (what does this mean?). If published, this will include your full peer review and any attached files.

Reviewer #1: **Yes: **Agmas Wassie Abate

Reviewer #2: No

---

## [Author Response · Author response to Decision Letter 1]

30 Aug 2023

Thank you Acadamic editor and both reviewers for this opportunity.

---

## [Editor Report · Decision Letter 2]

6 Sep 2023

PONE-D-22-34183R2POST-TRAUMATIC STRESS DISORDER AND ITS ASSOCIATED FACTORS AMONG WAR-AFFECTED RESIDENTS IN WOLDIA TOWN, NORTH EAST ETHIOPIA, 2022; COMMUNITY BASED CROSS-SECTIONAL STUDY.PLOS ONE

Dear Dr. kassaye,

Thank you for submitting your manuscript to PLOS ONE. After careful consideration, we feel that it has merit but does not fully meet PLOS ONE’s publication criteria as it currently stands. Therefore, we invite you to submit a revised version of the manuscript that addresses the points raised during the review process.

I regret to inform you that the revision was not carried out carefully. Therefore, you are requested to create a full revision for all and complete comments again.

We look forward to receiving your revised manuscript.

Kind regards,

Inga Schalinski

Academic Editor

PLOS ONE

Journal Requirements:

Additional Editor Comments:

Thanks you for including more information on the cumulative trauma score that you used in the study. However, this sentence needs revision in order to improve language:

“Cumulative trauma [20, 25], to assess cumulative exposure to traumatic event with the development of PTSD (dose-response association) was assessed by taking categories from different literatures, as exposure to single event, two to three events and four and above traumatic events.”

Suggestion: “Cumulative trauma [20, 25] was assessed to examine the dose-response association between exposure to traumatic events and the development of PTSD. Exposure categories were derived from various sources, including single events, two to three events, and four or more traumatic events.“

Please revise this once more if this score pertains to the frequency of exposure to the same traumatic event or to the exposure to different types of traumatic events.

Please discuss the deviance from a lot of findings form the literature in your discussion.

Your sentence has some minor grammatical issues:

„With regard to cumulative traumatic events, (43.4%) of respondents had experienced for two to three traumatic events and 92(15.4%) of238 participants had exposed for four and above traumatic events.“

Suggestion:

„In terms of cumulative traumatic events, 43.4% of respondents had experienced two to three traumatic events, and 92 (15.4%) out of 238 participants had been exposed to four or more traumatic events.“

For reviewer 2, I have found that you have deleted some parts of the comment. Please go back to the comments and present complete reviewers comments.

Response to comment 1: Please make clear how you address this issue in your paper. There seemed to be no changes made.

#2. The reviewer has kindly asked you to present M and SD of the PTSD severity score.

## 11 and # 18 please make your decision clear regarding the p-values.

Table 2. Please report the frequencies for each sex as requested by the reviewer despite presenting the general frequency. I would recommend to present the frequency for each sex in brackets in Table 2

Please respect the comment regarding Table 2.

---

## [Author Response · Author response to Decision Letter 2]

26 Sep 2023

This is a document with full and complete revision of all comments raised by the Academic Editor and Reviewers.

With Regard,

---

## [Editor Report · Decision Letter 3]

2 Oct 2023

POST-TRAUMATIC STRESS DISORDER AND ITS ASSOCIATED FACTORS AMONG WAR-AFFECTED RESIDENTS IN WOLDIA TOWN, NORTH EAST ETHIOPIA, 2022; COMMUNITY BASED CROSS-SECTIONAL STUDY.

PONE-D-22-34183R3

Dear Dr. kassaye,

We’re pleased to inform you that your manuscript has been judged scientifically suitable for publication and will be formally accepted for publication once it meets all outstanding technical requirements.

Kind regards,

Inga Schalinski

Academic Editor

PLOS ONE
---

## [Editor Report · Acceptance letter]

12 Dec 2023

PONE-D-22-34183R3 

POST-TRAUMATIC STRESS DISORDER AND ITS ASSOCIATED FACTORS AMONG WAR-AFFECTED RESIDENTS IN WOLDIA TOWN, NORTH EAST ETHIOPIA, 2022; COMMUNITY BASED CROSS-SECTIONAL STUDY. 

Dear Dr. kassaye:

I'm pleased to inform you that your manuscript has been deemed suitable for publication in PLOS ONE. Congratulations! Your manuscript is now with our production department. 

Kind regards, 

on behalf of

Dr. Inga Schali 

Academic Editor

PLOS ONE